# μPEN: Multi-class PseudoEdgeNet for PD-L1 assessment

**Jeroen Vermazeren**[1]                                          JEROEN.VERMAZEREN@RADBOUDUMC.NL
**Leander van Eekelen**[1]                                    LEANDER.VANEEKELEN@RADBOUDUMC.NL
**Luca Meesters**[1]                                                     LUCA.MEESTERS@RADBOUDUMC.NL
**Monika Looijen-Salamon**[1]                MONIKA.LOOIJEN-SALAMON@RADBOUDUMC.NL
**Shoko Vos**[1]                                                                  SHOKO.VOS@RADBOUDUMC.NL
**Enrico Munari**[2]                                                                  ENRICO.MUNARI@UNIBS.IT
**Caner Mercan**[1]                                                        CANER.MERCAN@RADBOUDUMC.NL
**Francesco Ciompi**[1]                                        FRANCESCO.CIOMPI@RADBOUDUMC.NL

[1] *Radboud University Medical Center, Department of Pathology, Nijmegen, the Netherlands.*

[2] *Department of Molecular and Translational Medicine, University of Brescia, Brescia, Italy.*

## Abstract

In this paper, we take the recently presented PseudoEdgeNet model to the level of multi-class cell segmentation in histopathology images solely trained with point annotations. We tailor its loss function to the challenges of multi-class segmentation and equip it with an additional false positive loss term. We evaluate it on the assessment of tumor and immune cells in PD-L1 stained lung cancer histopathology images, and compare it with YOLOv5.

**Keywords:** PseudoEdgeNet, YOLO, PD-L1, lung cancer, histopathology.

## 1. Introduction

Cell detection and segmentation in histopathology images are core steps in research and development of digital biomarkers that rely on counting, quantifying and analyzing shape and spatial interaction of multiple cell types. One example is the tumor proportion score (TPS), defined as the fraction of tumor cells positive to a PD-L1 immunohistochemical staining over all tumor cells in a tumor biopsy, assessed by pathologists to select non-small cell lung cancer (NSCLC) patients to receive immunotherapy. Computer assisted assessment of the TPS can address the current limitations in the subjective interpretation of PD-L1 expression at cell level, such as the presence of other PD-L1 positive cells (e.g. macrophages, to be excluded from the TPS) or the estimation uncertainty inherent in assessing potentially hundreds of thousands of cells in a tissue sample. We propose to automate the cell quantification task at the core of patient selection by detecting, classifying and segmenting PD-L1 positive (PD-L1$^+$) and PD-L1 negative (PD-L1$^-$) tumor and immune cells in PD-L1 stained NSCLC tissue samples, a task that, to the best of our knowledge, has only been addressed in (Althammer et al., 2019) using a closed-source solution. We developed *μ-PseudoEdgeNet* (μPEN), a novel multi-class formulation of PseudoEdgeNet (PEN) (Yoo et al., 2019), to produce multi-class cell segmentation from cell point annotations. In addition to altering the loss terms for multi-class segmentation, we equip μPEN with a false-positive loss to promote specificity. We empirically show the contribution of each loss term and benchmark detection performance versus the state-of-the-art detection method YOLOv5[1].

1. doi:10.5281/zenodo.4418161

## 2. μ-PseudoEdgeNet

μPEN updates and expands (Figure 1, green area) PEN's segmentation network $f$, edge network $g$ and attention network $h$, to predict multi-class cell segmentation when only trained with point annotations in input patches $I$ using the following loss function $L$:

$$L(S,T) = \underbrace{-\frac{1}{\#P}\sum_{i \in P}\sum_{c>0}log(S_i^c T_i^c) - \frac{1}{\#V}\sum_{i \in V}log(S_i^{c=0})}_{Segmentation\ loss} + \underbrace{\lambda_1 \frac{1}{\#F}\sum_i |s(F) - g(I)h(I)|}_{Edge\ loss} - \underbrace{\lambda_2 \frac{1}{\#T^{c=0}}\sum_{i \in B_{fp}}log(S_i^{c=0})}_{False\ positive\ loss},$$

where $S_i^c$ is the softmax output at pixel $i$ and class $c$, $T_i^c$ is the point annotations in the ground truth matrix. To compute the *segmentation loss*, we first calculate the Voronoi boundaries $V$ from all point annotations $P$, from which we compute the cross-entropy with non-background predictions $S^{c>0}$. Similarly, we compute the cross-entropy with the background predictions $S^{c=0}$ for all pixels on Voronoi boundaries $V$. To ensure cell boundary segmentation, the *edge loss* is calculated as the sum of the absolute difference between the output of the Sobel filter $s$ and the element-

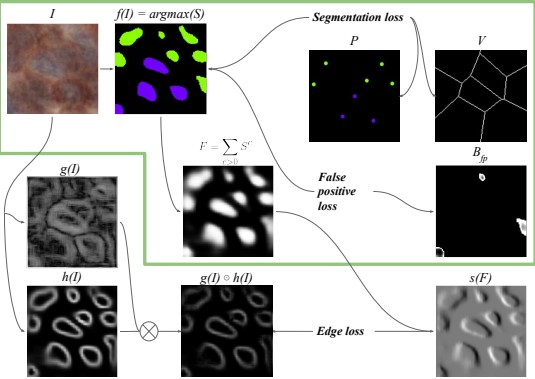

Figure 1: μ-PseudoEdgeNet.

wise multiplication of the output of the edge network $g$ and the attention module $h$, similar to the original PEN. In our multi-class version, we adapt the filter to be applied on the sum of foreground predictions, $F = \sum_{c>0} S^c$. Inspired by (Laradji et al., 2018), we introduced a *false positive loss* term by computing the cross-entropy of all connected components that cannot be associated with any point annotations in $P$. This loss term penalizes non-background predictions in false positive connected components denoted by $B_{fp}$, and we weigh its contribution by introducing the scaling constants $\lambda_1$ and $\lambda_2$, which are set to 1 and 10 after empirical evaluation. We train the same CNN backbones as in the original PEN from scratch with He initialization. At test time, we take the majority vote of all classes in a connected component and apply morphology based post-processing and test time color and shape augmentations. As a benchmark for μPEN, we also apply YOLOv5 to our dataset, using the default setting of its smallest model (YOLOv5s, 7.3M parameters, release 4.0), pretrained on the MS COCO dataset, only increasing the training IoU threshold to 0.25.

## 3. Experimental results

We collected n=39 whole-slide NSCLC histopathology images from 33 patients, stained for PD-L1 and digitized at 40× magnification. A trained medical research assistant (LM) supervised by lung pathologists (MLS, SV) selected 87 regions of interest (ROI) of 250×250 $\mu$m and manually annotated 32,180 cells in total with point annotations of tumor and immune cells, either PD-L1$^+$ or PD-L1$^-$ (four classes in total); annotations where checked by pathologists. We applied a data split of 21/9/9 for training/validation/testing, balanced at both patient and cell class level across sets. For YOLOv5, all point annotations were extended to bounding boxes of size 10×10 $\mu$m.

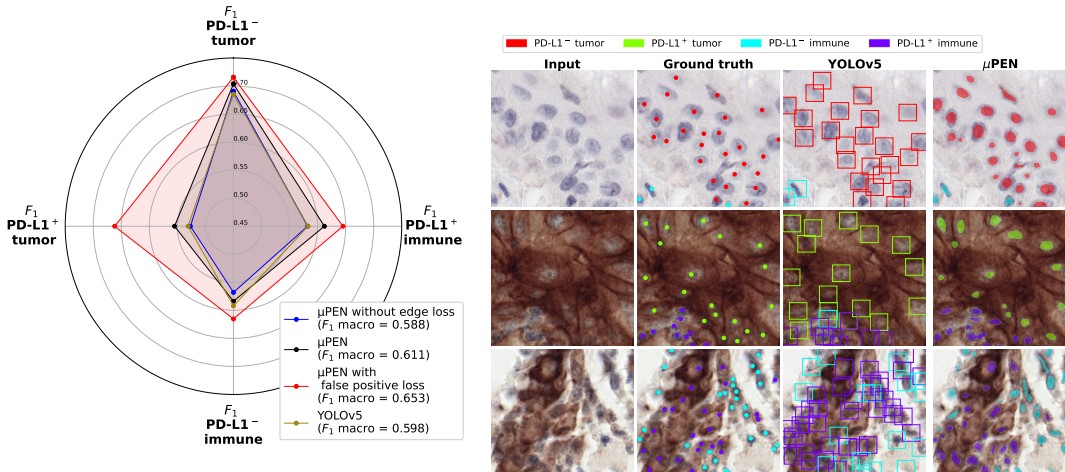

Figure 2: (Left) Polar chart showing the F1 scores per class & macro-averages. $\mu$PEN without FP loss is functionally equivalent to PEN. (Right) Visual results for YOLOv5/$\mu$PEN.

We trained $\mu$PEN and YOLOv5 on 512×512px patches ($0.25\mu$m/px) and selected best models based on the lowest validation loss. We translated predictions to points by taking the center-of-mass of the segmentation ($\mu$PEN) or bounding box (YOLOv5) and compared models' performance via the F1 score: an annotation is a hit when a detection is within a $4\mu$m distance (average radius of a cell in our dataset). Figure 2 depicts the incremental improvement of $\mu$PEN over PEN using the proposed false positive loss terms (0.611 vs. 0.653). It also shows that $\mu$PEN outperforms YOLOv5 when used "off-the-shelf". However, further hyper-parameter tuning could boost YOLOv5's performance.

## 4. Conclusions

The output of $\mu$PEN can potentially power (semi-)automated TPS assessment via cell localization and classification, as well as future biomarker research based on spatial interaction, size and morphology of different cell types without the need to train with manual annotations of cell borders. This multi-class framework can be easily extended to include non-tumor and non-immune cells, making this approach applicable to whole-slide images.

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
