# OpenReview forum: "$\mu$PEN: Multi-class PseudoEdgeNet for PD-L1 assessment"
_MIDL.io/2021/Conference/Short — MIDL 2021 Poster_

### Official Review · Reviewer_awJ2 · 2021-04-30

**Confidence:** 5
**Final Rating:** 3

**Summary:**

The authors present an extension of the PseudoEdgeNet (PEN) to obtain a trained model for image segmentation based on data-points annotations. They update the term of the loss function that deals with the detection task by including a false positive counting.
They compare their work with a pretrained YOLOv5.

**Strengths:**

The authors compare their proposal with the YALOv5, showing the differences between this approach and a different one.
The description of the new loss function is well written.
The numerical data and the visualization of the results are nicely presented.


**Weaknesses:**

As they are introducing an update of the existing PEN, I think they should compare their work with that one, rather than with YOLOv5, so they can show whether such changes improve the performance of the training.

A pretrained YOLOv5 was used in this case. What about the $\mu$-PEN? Was it also used with some pretrained weights?

**Deanonymize Review:**

no

**Justification Of The Rating:**

The novelty of the work is OK to be abstract. However, I think the method chosen for comparison is no the proper one. There is no further contribution besides an updated loss function for which the code to use it is not provided either.

**Paper Type:**

validation/application paper

**Special Issue:**

no

---

### Official Review · Reviewer_XYU8 · 2021-05-04

**Confidence:** 5
**Final Rating:** 3

**Summary:**

The paper presents a method for the automatic assessment of tumor and immune cells in PD-L1 stained lung cancer histopathology images. The deep learning-based method that is presented in this paper is composed of three different losses, one segmentation loss, one loss that takes into account the edges produced by a Sobel filter, and a false positive loss. The framework was trained by using only point annotations for 4 different cell classes.


**Strengths:**

- An interesting and easy-to-follow paper.
- A framework to provide multiclass segmentation of nuclei using only annotation of points.
- The proposed method reports better performance than YOLOv5 framework, while an ablation study is presented to highlight the need for the different components of the framework.


**Weaknesses:**

- There is no discussion about the computational/ time complexity of the provided framework.
- I miss a comparison with other weakly supervised methods similar to [1].

[1] Laradji, I.H., Rostamzadeh, N., Pinheiro, P.O., Vazquez, D., Schmidt, M.: Whereare the blobs: Counting by localization with point supervision. In: Proceedings ofthe European Conference on Computer Vision (ECCV). pp. 547–562 (2018)


**Deanonymize Review:**

no

**Detailed Comments:**

-What is the magnification that \muPEN was trained and tested?
- What is the time needed for a whole slide prediction of \muPEN?


**Justification Of The Rating:**

Justification Of The Rating: Overall I think the paper is well written, easy to follow, and provides an interesting contribution for MIDL 2021. I therefore recommend a rating of weak accept for this paper.


**Paper Type:**

methodological development

**Special Issue:**

no

---

### Meta-Review · Program_Chairs · 2021-05-06

**Recommendation:** Accept (Poster)
**Confidence:** 5

**Metareview:**

Both reviewers support acceptance. Authors are suggested to address reviewer comments in final version.

---

### Decision · Program_Chairs · 2021-05-11

Accept (Poster)